# Cortical D1 and D2 dopamine receptor availability modulate methylphenidate-induced changes in brain activity and functional connectivity

Peter Manza [1✉], Ehsan Shokri-Kojori [1], Şükrü Barış Demiral[1], Corinde E. Wiers[1], Rui Zhang[1], Natasha Giddens[1], Katherine McPherson [1], Erin Biesecker[1], Evan Dennis[1], Allison Johnson[1], Dardo Tomasi [1], Gene-Jack Wang [1] & Nora D. Volkow[1✉]

Dopamine signaling plays a critical role in shaping brain functional network organization and behavior. Prominent theories suggest the relative expression of D1- to D2-like dopamine receptors shapes excitatory versus inhibitory signaling, with broad consequences for cognition. Yet it remains unknown how the balance between cortical D1R versus D2R signaling coordinates the activity and connectivity of functional networks in the human brain. To address this, we collected three PET scans and two fMRI scans in 36 healthy adults (13 female/23 male; average age 43 ± 12 years), including a baseline D1R PET scan and two sets of D2R PET scans and fMRI scans following administration of either 60 mg oral methylphenidate or placebo (two separate days, blinded, order counterbalanced). The drug challenge allowed us to assess how pharmacologically boosting dopamine levels alters network organization and behavior in association with D1R-D2R ratios across the brain. We found that the relative D1R-D2R ratio was significantly greater in high-level association cortices than in sensorimotor cortices. After stimulation with methylphenidate compared to placebo, brain activity (as indexed by the fractional amplitude of low frequency fluctuations) increased in association cortices and decreased in sensorimotor cortices. Further, within-network resting state functional connectivity strength decreased more in sensorimotor than association cortices following methylphenidate. Finally, in association but not sensorimotor cortices, the relative D1R-D2R ratio (but not the relative availability of D1R or D2R alone) was positively correlated with spatial working memory performance, and negatively correlated with age. Together, these data provide a framework for how dopamine-boosting drugs like methylphenidate alter brain function, whereby regions with relatively higher inhibitory D2R (i.e., sensorimotor cortices) tend to have greater decreases in brain activity and connectivity compared to regions with relatively higher excitatory D1R (i.e., association cortices). They also support the importance of a balanced interaction between D1R and D2R in association cortices for cognitive function and its degradation with aging.

---

[1] National Institute on Alcohol Abuse and Alcoholism, National Institutes of Health, Bethesda, MD, USA. ✉email: peter.manza@nih.gov; nvolkow@nida.nih.gov

The neurotransmitter dopamine modulates the function of widespread brain networks, influencing a diverse range of behaviors including reward, motivation, motor, and cognitive function[1]. Dopamine exerts influence through two main types of receptors, the D1-like and the D2-like receptors (D1R/ D2R), which activate and inhibit adenylate cyclase, respectively, producing opposing effects on neuronal firing and fine tuning each others actions[2]. Yet it remains unclear how the balance between dopaminergic D1R and D2Rs signaling in the cortex impacts human brain function. The most widely used tool for studying human brain function relies on fMRI to measure blood-oxygenation level dependent (BOLD) signals as an index of brain activity and functional connectivity[3]. A better understanding of how dopamine signaling systematically influences the BOLD signal would benefit research on normal brain function as well as research on conditions with abnormal dopaminergic signaling, including substance use disorders, attention deficit/hyperactivity disorder, schizophrenia, and Parkinson's disease.

Early studies found that task-evoked dopamine release increased striatal BOLD signals[4,5], but could not speak to how D1R/D2R signaling contributed to the results. Further, although midbrain dopamine neurons have direct projections to striatum, their signals cascade throughout the brain[6,7] based on topographically-organized cortico-striatal 'loops'[8]. These larger patterns of signaling play a fundamental role in shaping healthy behavior, especially cognition[9], and have been largely understudied. Most studies in this space have used drug challenges to manipulate DA receptor signaling while observing changes in fMRI signals (i.e., pharmacological fMRI), or correlated static PET measures of dopamine receptor availability with fMRI[10–14]. While these studies provided important insights, it is difficult to identify an overarching pattern of how D1R and D2R signaling in cortical and subcortical regions impacts brain function, because these studies tended to focus on specific seed regions that differed across studies. In addition, measuring receptor availability is only one part of the picture; it remains unknown how dopamine increases (e.g., from pharmacological manipulation) would affect brain activity and connectivity based on D1R and D2R availability and to their relative expression (i.e., the D1R/D2R ratio) across the brain. Some of the best evidence to date comes from non-human primate studies. Using the knowledge that D1Rs are lower-affinity than D2Rs and only stimulated at high dopamine levels, Mandeville et al. found that high versus low doses of amphetamine produced increases and decreases in the striatal fMRI signal, respectively. These results informed a model whereby a high relative occupancy of D1Rs to D2Rs should produce increases in the fMRI signal, and vice versa.

We sought to test the predictions of this model in humans and explain dopamine-induced changes in brain activity and connectivity across the neocortex. To do this, we collected baseline measures of D1R and D2R availability using PET and resting functional connectivity using fMRI; we also administered methylphenidate (60 mg oral) prior to a second D2R PET scan and fMRI sessions (Fig. 1). Based on the model of Mandeville et al., we hypothesized that networks with a higher relative D1R/ D2R ratio (and presumably a greater capacity for excitatory responses to dopaminergic stimulation) would show increases in brain activity to the methylphenidate challenge. Conversely, networks with lower D1R/D2R ratios would show decreases in activity to methylphenidate. We also expected that changes in within-network functional connectivity would parallel the changes in activity, since studies utilizing both glucose metabolism and fMRI have found strong correspondence between local activity and functional connectivity[15–17]. Finally, due to a broad literature on the topic, we tested how relative D1R/D2R ratios were associated with age and with cognitive performance focusing on

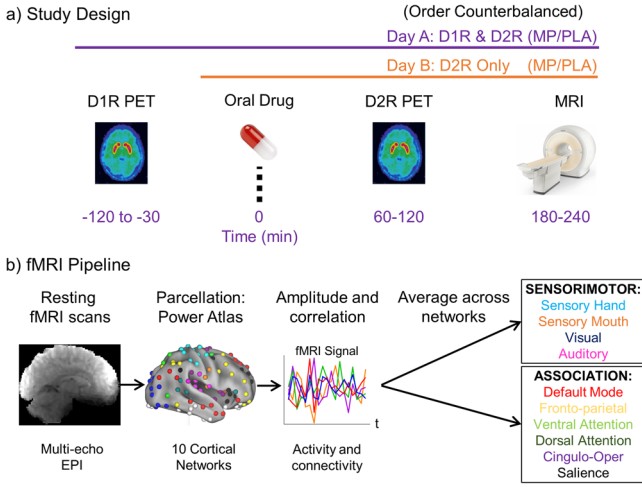

**Fig. 1 Study design and fMRI Pipeline. a** Study design. On one day, participants underwent a [¹¹C]NNC-112 scan to assess D1R availability at baseline. Then, participants took the oral medication (either 60 mg methylphenidate, MP, or placebo, PLA) and underwent a [¹¹C]raclopride scan to assess D2R availability, and a resting fMRI scan to assess brain activity and connectivity. On a second day, the D2R and fMRI scans were repeated after the other oral drug was given (MP/PLA session order was counterbalanced). The second set of fMRI scans allowed us to estimate changes in brain activity and connectivity from MP-induced dopamine increases, relative to placebo. **b** fMRI pipeline. Multi-echo echo-planar imaging (EPI) scans were acquired and preprocessed (see Methods section). Then, signals were extracted from 10 cortical networks of the Power et al. atlas, and the average amplitude and within-network correlations were computed to estimate activity and connectivity, respectively. Finally, we averaged activity and connectivity estimates across the four sensorimotor networks and the six association networks. Brain image under 'Parcellation: Power Atlas' is reprinted from Neuron, 72(4), Power, J.D., Cohen, A.L., Nelson, S.M., Wig, G.S., Barnes, K.A., Church, J.A., Vogel, A.C., Laumann, T.O., Miezin, F.M., Schlaggar, B.L. and Petersen, S.E, "Functional network organization of the human brain", Pages 665–678, Copyright (2011), with permission from Elsevier.

spatial working memory. Based on consistent preclinical evidence that dopamine receptor signaling in prefrontal and parietal cortex is critical for spatial working memory[18], we hypothesized that the D1R/D2R ratio in the association (as opposed to sensorimotor) cortices would be especially relevant for cognition and would decrease with aging.

## Results

We first mapped the relative availability of D1R, D2R, and the D1R/D2R ratio across the neocortex, and tested if these measures significantly differed across sensorimotor versus association cortices, using paired $t$-tests (map of the D1R/D2R ratio is shown in Fig. 2a). There were robust differences across canonical functional brain networks (Fig. 2b), such that relative D1R availability ($t_{(35)} = 46.34$, $p < 0.0001$) and relative D2R availability ($t_{(35)} = 32.25$, $p < 0.0001$) were much greater in association as compared to sensorimotor cortices. However, the regional differences were greater in D1R than D2R availability, which showed as a significantly higher relative D1R/D2R ratio in association as compared to sensorimotor cortices $t_{(35)} = 15.07$, $p < 0.0001$; Fig. 2c). All relative PET measures met the assumption of a normal distribution (Kolmogorov–Smirnov distances < 0.12; all $p$'s > 0.10). Inspection of individual networks revealed that D1R and D2R availability was highest in the cingulo-opercular network and lowest in the sensory hand network; meanwhile, the

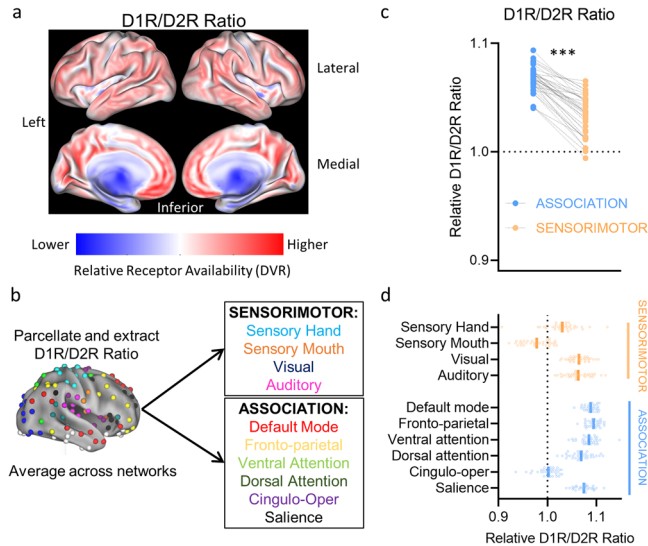

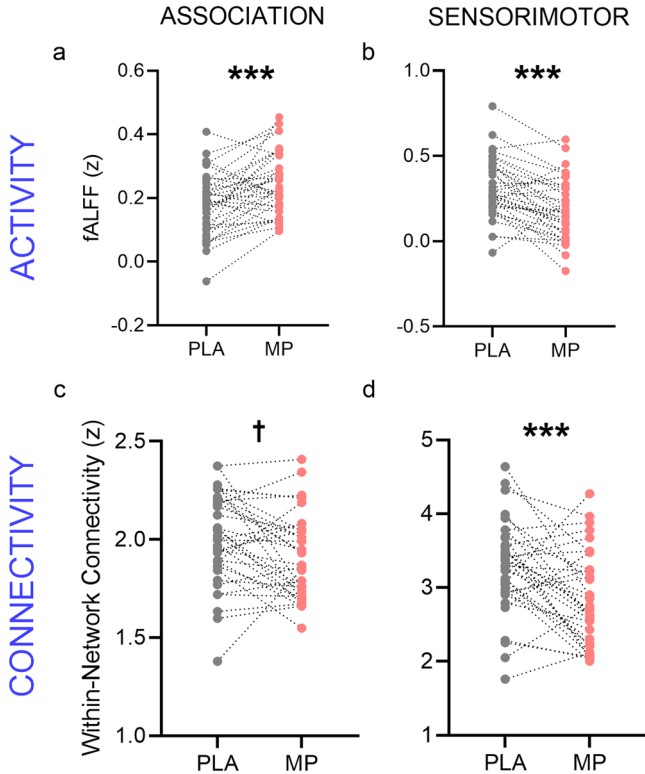

**Fig. 2 Relative D1-D2 Ratio across the neocortex. a** Relative (intensity-normalized) map of the ratio of dopamine D1-D2 receptor availability (relative D1R/D2R ratio). **b** Relative D1R/D2R ratio was extracted for each region of a 10-network parcellation, and averaged across all regions of each network. **c** The relative D1R/D2R ratio, for the average of all association and all sensorimotor networks. **d** The relative D1R/D2R ratio by individual networks (for visualization only). In panel **c** and **d**, each dot reflects a participant. In panel **d**, the solid lines represent the group mean for each network. Note: DVR = distribution volume ratios. ***$p < 0.001$. Brain image in panel **b** is reprinted from Neuron, 72(4), Power, J.D., Cohen, A.L., Nelson, S.M., Wig, G.S., Barnes, K.A., Church, J.A., Vogel, A.C., Laumann, T.O., Miezin, F.M., Schlaggar, B.L. and Petersen, S.E, "Functional network organization of the human brain", Pages 665–678, Copyright (2011), with permission from Elsevier.

**Fig. 3 Methlyphenidate-induced changes in brain activity and connectivity across association and sensorimotor cortices.** Methylphenidate (MP)-induced changes in brain activity (**a**, **b**) and connectivity (**c**, **d**), relative to placebo (PLA). fALFF fractional amplitude of low-frequency fluctuations. ***$p < 0.001$; †$p = 0.05$.

relative D1R-D2R ratio was highest in the fronto-parietal network and lowest in the sensory mouth network (Fig. 2d). Since [11C]Raclopride has lower sensitivity to detect extrastriatal D2R than other radiotracers like [18F]Fallypride[19], we conducted a secondary analysis to see if our results with [11C]Raclopride generally agree with those of [18F]Fallypride. We performed a voxelwise spatial correlation of D2R availability in the neocortex between the group average Raclopride map in the current study and the group average [18F]Fallypride map of 25 healthy adults from a publicly availably atlas[20]. There was moderate-to-strong correspondence (linear $r = 0.484$, quaratic $r = 0.759$; see Supplementary Fig. 1), and results from both radiotracers showed higher D2R availability in association as compared to sensorimotor cortices.

We then tested how increases in dopamine would change brain activity and connectivity in sensorimotor and association cortices. Paired $t$-tests (methylphenidate versus placebo) indicated opposing changes in brain activity across regions: methylphenidate significantly increased the fractional amplitude of low-frequency fluctuations (fALFF) in association regions ($t_{(34)} = 3.895$, Bonferroni-corrected $p = 0.0008$; Fig. 3a) whereas it significantly decreased fALFF in sensorimotor regions ($t_{(34)} = 4.689$, Bonferroni-corrected $p < 0.0001$; Fig. 3b). Likewise, the patterns of functional connectivity differed across regions: methylphenidate did not produce robust differences in within-network connectivity in association regions ($t_{(34)} = 2.365$, Bonferroni-corrected $p = 0.05$; Fig. 3c) but it significantly decreased within-network connectivity in sensorimotor regions ($t_{(34)} = 3.779$, Bonferroni-corrected $p = 0.0012$; Fig. 3d). ANOVAs testing for network-by-medication interactions yielded significant interaction effects (for fALFF: $F_{(1,34)} = 35.69$, $p = 9.32 \times 10^{-7}$, $\eta^2 G = 0.121$; for functional connectivity: $F_{(1,34)} = 11.33$, $p = 0.003$, $\eta^2 G = 0.035$).

We also examined whether non-relative striatal D1R and D2R availability, D1R/D2R ratio, and methylphenidate-induced dopamine increases were associated with changes in brain activity and connectivity. To replicate prior work, we confirmed that methylphenidate significantly decreased D2R availability in ventral ($t_{(34)} = 4.835$, $p < 0.0001$) and dorsal striatum ($t_{(34)} = 6.213$, $p < 0.0001$), reflecting dopamine increases. At an uncorrected threshold, D1R in dorsal and ventral striatum were positively correlated with baseline (placebo session) brain activity in sensorimotor cortices (for each dorsal and ventral striatum, $r^2 = 0.115$, $p = 0.043$); however these results would not survive correction for multiple comparisons. None of the other baseline PET measures significantly correlated with baseline measures of activity/connectivity in association nor sensorimotor cortices, even at an uncorrected threshold. Further, methylphenidate-induced changes in D2R availability (dopamine increases) in ventral and dorsal striatum did not significantly correlate with methylphenidate-induced changes in brain activity/connectivity.

Finally, we tested whether the relative D1R/D2R ratio was significantly associated with age and spatial working memory performance. As hypothesized, we showed a significant negative correlation with age in association ($r^2 = 0.206$, uncorrected $p = 0.006$; Fig. 4a) but not sensorimotor cortices ($r^2 = 0.013$, $p = 0.508$; Fig. 4b) such that there was an age-related decline in relative D1R/D2R ratio in the association cortices. Additionally we showed that the relative D1R/D2R ratio correlated positively with spatial working memory performance—that is, negatively correlated with task errors—in association ($r^2 = 0.143$, uncorrected $p = 0.027$; Fig. 4c) but not in sensorimotor cortices ($r^2 = 0.030$, $p = 0.330$; Fig. 4d) such that higher ratios were linked with better performance. Critically, neither the relative D1R nor

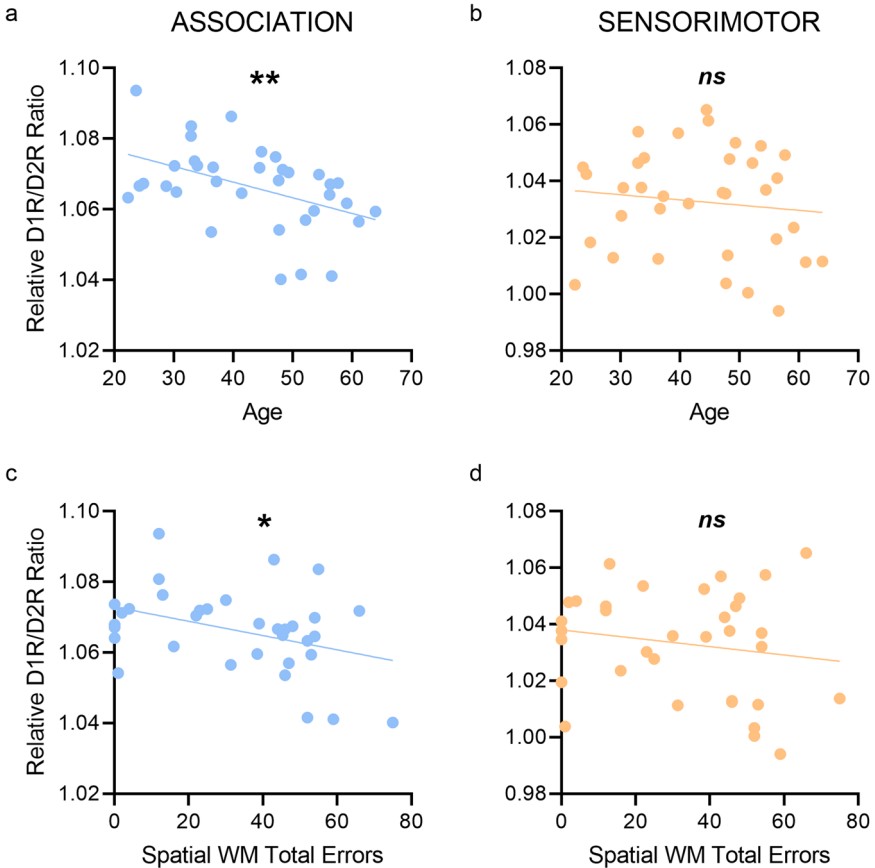

**Fig. 4 Associations of relative D1R/D2R ratio with age and spatial working memory performance.** Correlations between relative D1R/D2R ratio and age (**a**, **b**) and cognitive performance (**c**, **d**). Trend lines reflect the regression line of best fit. WM working memory. **$p < 0.01$; *$p < 0.05$; ns not significant ($p > 0.05$).

relative D2R availability on their own was significantly associated with age or SWM performance (all $p$'s > 0.05), suggesting that the relative D1R/D2R ratio was more sensitive than either receptor alone.

## Discussion
Using PET, fMRI, and a pharmacological challenge, we found evidence that methylphenidate-induced dopamine increases (and presumably also norepinephrine increases) triggered changes in brain function that differed between association and sensorimotor networks, and were associated with the relative availability of dopamine D1/D2Rs in these regions. Association networks, with a high relative concentration of excitatory D1Rs to inhibitory D2Rs, showed strong drug-induced increases in resting brain activity. Meanwhile, sensorimotor networks, with comparatively lower D1R/D2R ratios, showed decreases in resting brain activity and functional connectivity; these data largely confirmed predictions from an extant model[21]. Across individuals, a higher D1R/D2R ratio in association cortices was negatively correlated with age and positively correlated with spatial working memory performance. These data provide a possible framework for how medications like methylphenidate differentially affect resting functional networks, and highlight the importance of the relative balance of cortical D1R to D2R signaling in cognitive function and in brain aging.

Using data from a sample of healthy adults, we mapped the relative availability of D1R and D2R across the cortex, as previously done with PET measures of glucose metabolism using FDG[22,23]. Though [11C]NNC-112 has long been used to map D1Rs in the neocortex, only recently have studies found that

[11C]raclopride can also be used to observe a reliable D2R signal extrastriatally[24,25]. By taking advantage of this information, we were able to identify regional patterns of the two receptor densities and, critically, the relative signaling of D1R to D2R ratios. We found that, while both D1Rs and D2Rs show a much higher concentration in association as compared to sensorimotor cortices, the difference was much greater for D1Rs. This resulted in a much higher D1R/D2R ratio in association cortices than in sensorimotor cortices. To our knowledge this finding has not been described previously, but this may fit with human postmortem studies on D1R and D2R densities in different parts of the striatum, and what we know of these striatal subregions' anatomical projections. The head of caudate, which has strong reciprocal connections with association cortices including frontal and parietal cortices, has relatively higher D1R binding and lower D2R binding than the posterior putamen, which is highly interconnected with somatosensory cortices[8,26,27]. If the D1R/D2R ratio indeed follows these topographic cortico-striatal connections, we speculate that the high relative prevalence of D1Rs in association cortices has especial relevance for cognition: for instance, Alzheimer's disease, marked by aberrant brain function in predominantly association cortices (including the default mode network; refs. [28,29]) is associated with a reduction in D1R but not in D2R availability, relative to controls[30].

We hypothesized that, if D1R signaling is predominantly excitatory and D2R signaling inhibitory[2], then dopamine increases would increase activity/connectivity in regions with higher D1R/D2R ratios, whereas regions with lower D1R/D2R ratios would decrease activity/connectivity. To test this, we administered a high dose of oral methylphenidate and found that

our hypotheses were largely confirmed: resting fMRI activity (measured with fALFF) strongly increased after methylphenidate compared to placebo in association cortices (with higher D1R/D2R ratios), but strongly decreased in sensorimotor cortices (with lower D1R/D2R ratios). These data bear out in humans the predictions of models for how D1-like and D2-like signaling affects neurovascular responses in non-human primates[21]. They also broadly align with studies showing that D1R binding in amygdala positively correlated with BOLD responses in the amygdala to fearful faces[31], and with findings that D2R antagonists increase hemodynamic responses more in regions with higher D2R density[32].

Within-network connectivity followed a similar, though not identical pattern: connectivity also significantly decreased in sensorimotor cortices but a much weaker decrease in association cortices. These findings largely confirmed our hypothesis and correspond well with prior FDG/fMRI studies which found a strong coupling between local activity and functional connectivity[15–17]. The methylphenidate-induced decreases in sensorimotor within-network connectivity replicate prior work[33,34] and might be relevant to its therapeutic benefit: in adults with attention-deficit/hyperactivity disorder (ADHD), symptoms of hyperactivity and restlessness are associated with higher functional connectivity in these circuits[35]. Of note, methylphenidate also boosts central norepinephrine in addition to dopamine, raising the possibility that these findings could be due to increases in norephinephrine. However, a study using atomoxetine, a selective norepinephrine transporter blocker, found a somewhat different pattern from that in the current study: in response to an oral 40 mg dose of atomoxetine, functional connectivity tended to decrease across the brain along a posterior-to-anterior gradient, decreasing most in regions with high connectivity at baseline[36]. The different findings should be interpreted with caution, since some of the methodology differs between studies; nevertheless, it appears that methylphenidate and atomoxetine produce fairly different effects on regional brain function.

Lastly, the relative D1R/D2R ratio in association cortices negatively correlated with age and positively correlated with spatial working memory performance across individuals. It is well-documented that various markers of dopaminergic function decline with age[37–39]. Although one human postmortem study suggested that striatal D1Rs and D2Rs appear to decline with age at a roughly equal rate, leaving the ratio unchanged[40] others found that D1Rs decline with age more rapidly than D2Rs[41], and a recent meta-analysis of PET studies confirmed a more rapid decline in D1Rs (roughly 13–14% per decade) than D2Rs (roughly 8–9% per decade), both in frontal cortex and striatum[42]. Here we replicate the finding of a declining D1R/D2R ratio with age and find significant effects in association (including prefrontal) cortices, but not in sensorimotor cortices. Age-related decline in dopamine receptors has been linked to reductions in prefrontal metabolism[43] and cognitive performance[44,45]. Further, prefrontal D1R and D2R signaling have each been specifically implicated in spatial working memory performance[18,46,47]. Here we highlight that the relative ratio of these two receptors may be a critical marker for understanding cognitive function throughout the lifespan. However, these data are cross-sectional, thus we cannot determine if age-related decline in the D1R/D2R ratio causes the lower spatial working memory performance.

There were several limitations to the current study. PET data were collected on two different scanners; however, we used methods to correct for any average differences in the PET images based on scanner, and more importantly, the primary analyses in this manuscript were conducted within-subjects. Other limitations include the radiotracers used. Specifically, for [11C]NNC-112, though the signal is predominantly due to D1R, roughly 20%

of the cortical signal may reflect binding to serotonin 5HT2a receptors[48]. More specific radiotracers are needed to obviate this confound. In the case of [11C]raclopride, while it predominantly binds to D2-type receptors, its affinity is relatively low, thus specific to nonspecific signals from cortical regions are lower than other radiotracers such as [11C]fallypride[19,49]. Still, consistent evidence suggests that a reliable and detectable specific signal exists[50] and our work follows a number of recent investigations that have successfully used [11C]raclopride to study extrastriatal D2R[51–53]. Nevertheless, due to the somewhat lower specific signal of [11C]raclopride, the individual difference findings should be interpreted with some caution and viewed as preliminary data in need of replication. Additionally, while many prior studies have argued that fALFF is a good proxy for brain activity, the exact significance of fALFF remains unknown. Studies using concurrent electrical recording of neuronal activity and fMRI in monkeys and rodents have found that low-frequency BOLD fluctuations are strongly associated with gamma-band local field potential activity and to a lesser extent multiple-unit neuronal spiking activity[54–56]. Nevertheless, the precise relationship between BOLD amplitude and neuronal activity remains under debate. Therefore, the interpretation of fALFF as reflecting 'activity' should be done cautiously.

Future studies could expand on this work by utilizing simultaneous PET-pharmacological fMRI[57]. There are notable advantages to this approach. Compared to sequential acquisition of PET and fMRI images, simultaneous measurement would likely yield a stronger relationship between pharmacologically-induced changes in PET and fMRI measures, since it would eliminate temporal differences in drug bioavailability. Further, it would eliminate temporal differences in an individual's mental/physical state, which is especially relevant for fMRI, since some non-trivial component of resting fMRI measures appears to be state-based (studies estimate that the intra-class coefficient for resting fMRI measures varies between 0.5 and 0.8[58,59]). Finally, simultaneous acquisition allows one to leverage temporal patterns of PET and fMRI signals to model molecular adaptations to drugs such as receptor internalization, as recently performed in the non-human primate[60]. These efforts hold the promise of gaining richer insight into how dopamine receptor function shapes brain activity and connectivity.

In conclusion, we observed that relative availability of D1R and D2Rs and D1R/D2R ratio systematically varied across canonical sets of resting state networks. Methylphenidate produced a pattern of changes in brain activity and connectivity in a regional pattern aligning with these receptor densities, which was predicted by extant models of dopamine receptor signaling and fMRI responses[21]. Specifically, association cortices with higher relative D1R/D2R ratios, which presumably have greater potential for excitatory responses upon dopaminergic stimulation, tended to have greater increases in activity (and weaker decreases in connectivity) after a methylphenidate challenge, as compared to sensorimotor cortices with lower D1R/D2R ratios where methylphenidate decreased activity and connectivity. The D1R/D2R ratio in association cortices also seems to have relevance for aging and cognitive function, which highlights the potential of studying these systems to understand neuropsychiatric disorders including substance use, attention-deficit/hyperactivity disorder and Parkinson's disease[61,62].

## Methods

**Participants**. Data from 36 healthy adults were included in the study (23 male, 13 female, age 22–64; for participant characteristics see Table 1). All participants provided written informed consent, and the Institutional Review Board committee of the National Institutes of Health approved the study. Participants were excluded if they had a history of substance abuse or dependence (other than nicotine) or a

**Table 1 Demographics and participant characteristics, by PET scanner.**

|  | Scanner 1: HRRT (n = 17) | Scanner 2: PET/CT (n = 19) | $t_{(df)}$, p |
|---|---|---|---|
| **Age** | | | |
| Min-Max | 33–64 | 22–60 | – |
| Mean ± SD | 48.41 ± 9.60 | 39.31 ± 12.83 | $2.39_{(34)}$, 0.023 |
| **Sex** | | | |
| n, Female (%) | 7 (41) | 6 (32) | $0.062_{(1)}$, 0.801[a] |
| **BMI** | | | |
| Min-Max | 21–39 | 21–33 | – |
| Mean ± SD | 27.19 ± 5.09 | 28.24 ± 3.28 | $0.74_{(34)}$, 0.463 |
| **IQ** | | | |
| Min-Max | 79–139 | 97–129 | – |
| Mean ± SD | 122.35 ± 15.80 | 110.05 ± 11.58 | $2.64_{(34)}$, 0.011 |
| **Race** | | | |
| n, White (%) | 11 (65) | 5 (26) | $7.16_{(3)}$, 0.067[a] |
| n, Black/AA (%) | 6 (35) | 10 (53) | – |
| n, Asian (%) | 0 (0) | 2 (11) | – |
| n, Other (%) | 0 (0) | 2 (11) | – |

BMI body-mass index, IQ intelligence quotient, SD standard deviation.
[a]$\chi^2$ test-statistic.

history of psychiatric disorder, neurological disease, medical conditions that may alter cerebral function (i.e., cardiovascular, endocrinological, oncological, or autoimmune diseases), current use of prescribed or over-the-counter medications, and/or head trauma with loss of consciousness of >30 min. For an overview of the study flow, see Fig. 1a.

*MRI acquisition.* An overview of the fMRI pipeline is depicted in Fig. 1b. All subjects underwent structural and resting-state functional MRI on a 3.0T Magnetom Prisma scanner (Siemens Medical Solutions USA, Inc., Malvern, PA) with a 32-channel head coil. To acquire resting fMRI time series a multi-echo, multiband EPI sequence was used: multiband factor = 3, anterior-posterior phase encoding, TR = 891 ms, echo times = 16, 33, and 48 ms, flip angle = 57 deg, 45 slices with 2.9 × 2.9 × 3.0 mm voxels and 520 time points while the participant relaxed with their eyes open (total acquisition time = 8 min). A fixation cross was presented on a black background under dimmed room lighting using a liquid-crystal display screen (BOLDscreen 32, Cambridge Research Systems; UK). The 3D MP-RAGE (TR/TE = 2400/2.24 ms, FA = 8 deg) and variable flip angle turbo spin-echo (Siemens SPACE; TR/TE = 3200/564 ms) pulse sequences were used to acquire high-resolution anatomical brain images with 0.8 mm isotropic voxels field-of-view (FOV) = 240 × 256 mm, matrix = 300 × 320, and 208 sagittal slices.

*PET acquisition and drug administration.* PET scans were used to measure D1R availability with [11C]NNC-112 and to measure D2R availability with [11C]Raclopride. For each individual, studies were conducted on one of two scanners: a high-resolution research tomography (HRRT) scanner (n = 17; 7 female; Siemens AG; Germany) or a Biograph PET/CT scanner (n = 19; 6 females; Siemens AG; Germany). The use of two different scanners was necessary due to scheduling limitations at our site. The methods for correcting differences between scanners are described in the PET analysis section below. All [11C]NNC-112 scans were conducted at 10AM, in a baseline state, without any drug manipulation. [11C]raclopride scans were conducted on two separate days: once 1 h after administration of an oral placebo pill (to assess baseline D2R availability) and once 1 h after administration of 60 mg oral methylphenidate (single blind; counterbalanced session order). Raclopride scans were conducted at the same time of day (1 PM) and in the same scanner for a given participant.

For [11C]NNC-112, emission scans were started immediately after a maximum injection of 555 MBq. Twenty-one dynamic emission scans were obtained from time of injection up to 90 min after. For [11C]raclopride, emission scans were started immediately after a maximum injection of 370 MBq. Twenty-two dynamic emission scans were obtained from time of injection up to 60 min after. Dynamic emission scan images were evaluated before analyses to ensure that motion artifacts or misplacements were not included.

*PET analysis.* PET images were coregistered to the high-resolution MRI T1 and T2 structural images. We used the minimal preprocessing pipelines of the Human Connectome Project for the spatial normalization to the stereotactic MNI space of the structural and PET scans[63]. Differences in geometry and PSF between cameras (PET/CT = 4 mm PSF; HRRT = 2.7 mm PSF) resulted in systematic voxelwise differences in signal intensity between PET/CT and HRRT images. To correct for

these scanner-specific scaling effects and harmonize the data we used a voxelwise approach based on grand-mean scaling. We used an updated version of the ComBat Harmonization technique implemented in the ENIGMA study[64,65]. Originally proposed by Johnson et al. (2007) and implemented in the surrogate variable analysis (sva) package in R[66], ComBat uses an Empirical Bayes framework to estimate the distribution scanner effects. It was shown to be superior to other harmonization methods for varieties of data types including DTI[67] and cortical thickness[68]. We conducted ComBat separately for each tracer for the PET measure of interest (i.e., receptor availability) to harmonize the data across scanners. For [11C]raclopride measures, since we had placebo and methylphenidate treatments, we used drug, age, sex (male/female), and race (4 groups; white, black, Asian, and others) as covariates in the model. For [11C]NNC measures, since there was no drug manipulation, we used only age, sex, and race as covariates.

FreeSurfer version 5.3.0 (http://surfer.nmr.mgh.harvard.edu) was used to automatically segment the anatomical MRI scans using the Desikan atlas[69], which provided bilateral nucleus accumbens, caudate/putamen, and cerebellum regions of interest (ROIs).

**D1R/D2R availability: striatum.** Time–activity curves in the dorsal striatum (caudate and putamen), NAc, and cerebellum were used to obtain the distribution volume ratios using a Logan reference tissue model[70,71]. The accumbens-to-cerebellum and the dorsal striatum-to-cerebellum distribution volume ratios correspond to BPnd+1, which was used to quantify D1R and D2R receptor availability and the ratio of the availability of D1R to D2R (D1R/D2R). We averaged the values for caudate and putamen to create one 'dorsal striatum' ROI, since caudate and putamen BPnd are highly correlated with one another (across participants, $r \approx 0.9$). We also used the D2R availability estimates to compute 'dopamine increases' based on previous work:

$$\text{Dopamine Increases} = \frac{\text{D2Rplacebo} - \text{D2Rmethylphenidate}}{\text{D2Rplacebo}} \quad (1)$$

**Relative D1R/D2R availability: neocortex.** To assess regional differences in D1R/D2R availability, we computed relative images by normalizing the D1R/D2R DISTRIBUTION VOLUME RATIO maps to the whole brain mean (FSL's MNI_T1_2mm_brain_mask image). D1R/D2R availability in the neocortex as measured by PET is dominated by a global signal with very little regional variability: across participants the correlation between receptor availability in different cortical networks is very high (e.g., ref. [13]). Since our measures of brain function with fMRI (activity and connectivity) vary greatly across the networks, the relative measures of D1R/D2R availability enhance regional differences and are better suited for comparison with regional fMRI measures.

Note that, although [11C]Raclopride has been traditionally used only to measure striatal dopamine receptor availability due to lower D2R estimates in neocortex, recent work has found that it also produces reliable measures of exstrastriatal D2R[24,25]. Nevertheless, this remains under some debate and since [11C]Raclopride does have lower sensitivity to detect extrastriatal D2R than some other radiotracers[49], we took several precautions in the current study. First, we used spatial correlation to compare D2R availability in the neocortex between the group average Raclopride map in the current study and the group average [18F]Fallypride map of 25 healthy adults from a publicly availably atlas[20], since Fallypride has higher extrastriatal D2R sensitivity[19], and demonstrated moderate-to-strong correspondence (see Supplementary Fig. 1). Second, we limited our analysis to large-scale sets of networks (sensorimotor versus association cortices; see 'Brain network parcellation scheme' section below) to reduce the possibility of insufficient signal-to-noise ratio when examining small individual regions with Raclopride. Finally, we only examined estimates of receptor availability and did not attempt to examine changes in receptor availability in neocortex to the methylphenidate challenge (i.e., dopamine increases), since drug-induced changes in Raclopride binding may not be reliably detected in some regions of neocortex[19].

**Brain network parcellation scheme: sensorimotor and association cortices.** For resting fMRI network activity/connectivty and relative D1R/D2R maps, we followed procedures from prior work[22,72] and extracted data from broad sets of sensorimotor and association regions delineated using a popular resting fMRI parcellation scheme[73] (Fig. 1b). We used the consensus parcellation scheme (264 regions of interest) and created a 5 mm sphere at each region from which we extracted data. After computing each measure of interest (see processing sections below) we aggregated across networks by taking the average value across all sensorimotor and all association networks, following the precedent of prior studies[72]: Hand somato-motor, mouth somato-motor, visual, and auditory networks contributed to the sensorimotor system, and default mode, fronto-parietal, ventral attention, dorsal attention, cingulo-opercular, and salience networks contributed to the association system.

**Functional MRI processing: activity and connectivity.** For resting fMRI time series, the Human Connectome Project functional pipeline was used for gradient distortion correction, rigid body realignment, field map processing, and spatial

normalization to the stereotactic MNI space. 0.008-.09 Hz band-pass filtering was used to assess the low frequency fluctuations in the resting fMRI data. Signals from the white matter and CSF were regressed out of the data. Framewise displacements (FD) were computed from head translations and rotations using a 50 mm radius to convert angle rotations to displacements. Scrubbing was used to remove time points excessively contaminated with motion. Specifically, time points were excluded if the root mean square change in the BOLD signal (DVARS) from volume to volume met the criteria: DVARS > 0.5% and FD > 0.5 mm. For a measure of activity, we used the fractional amplitude of low-frequency fluctuations (fALFF), which maps spontaneous fluctuations in the 0.01–0.10 Hz frequency band. For a measure of connectivity, we used standard measures of within-network functional connectivity: Pearson correlation coefficients were calculated between pairs of ROI-averaged time courses after 0.01–0.10 Hz bandpass filtering and normalized to z-scores using the Fisher transformation.

**Cogniton: spatial working memory performance**. To assess cognitive performance, we used the spatial working memory task from the Cambridge Neuropsychological Test Automated Battery suite[74]. Participants completed this task in a baseline state, i.e. without having received a dose of methylphenidate or placebo. In this task, the participant sees an arrangement of colored squares on the screen. The computer hides a yellow token in one of the squares, and the participant must find it. Once the participant finds the token, the computer will hide it in another square, and it will never use the same square twice. There is a black bar on the side of the screen, where the participant puts each token once they have found it. The task is complete when the participant has found all of the tokens and filled the black bar. The primary outcome measure of this task is total number of errors, where fewer errors indicates better spatial working memory performance.

**Statistics and reproducibility**. Analyses were performed in R version 3.6.2 and in GraphPad Prism version 8.0.1.

To test for regional differences in the PET measures (relative D1R, relative D2R, and relative D1R/D2R ratio), we perfomed paired $t$-tests (association versus sensorimotor).

To test for methylphenidate-induced changes in the fMRI measures (brain activity/connectivity), we perfomed paired $t$-tests (placebo vs. methylphenidate), each for association and sensorimotor cortices, and Bonferroni-corrected for two comparisons. Then, to see if the pattern of methylphenidate-induced changes in brain activity/connectivity differed by cortical regions, we performed a two-way repeated measures ANOVA, with drug (placebo versus methylphenidate) and network (association versus sensorimotor) as factors, and examined the interaction effect.

As a control analysis, we also examined traditional measures of striatal receptor availability. We tested for methylphenidate-induced changes in the D2R availability (i.e., dopamine increases) and tested whether D1R, D2R, D1/D2 ratio, and dopamine increases correlated with baseline fMRI activity and connectivity, as well as methylphenidate-induced changes in activity and connectivity, both in sensorimotor and association cortices, using Pearson correlation.

Finally, we tested if the relative D1R/D2R ratio, each for association and sensorimotor cortices, was significantly associated with age and spatial working memory performance, using linear regression in Graphpad Prism. We hypothesized significant associations would be observed in the association cortices, based on a large literature showing that dopamine receptor signaling in association regions such as prefrontal/parietal cortices is critical for spatial working memory[18]. As an exploratory analysis, we repeated these tests in regression models that included sex and IQ as factors, using the lm function in R. Based on these tests, sex and IQ did not appear to play a major role in the results and these additional models are presented in Supplementary Note 1.

Due to poor image quality, fMRI data from one participant was removed. After this, fMRI analyses were performed twice: once with the entire sample ($n = 35$) and once after we removed participants with high levels of motion during resting fMRI ($n = 5$ removed due to >15% of timepoints scrubbed, reminder of particpants: $n = 30$). Since general findings did not change we report results from the full ($n = 35$) sample here.

**Reporting summary**. Further information on research design is available in the Nature Research Reporting Summary linked to this article.

## Data availability

Summary data used to produce primary results are in a publicly available repository at: https://github.com/pmanza/Cortical-D1D2. Source data underlying figures are also available in Supplementary Data 1–2.

## Code availability

R scripts (R version 1.2.5019) used to produce primary results are in a publicly available repository[75] at: https://github.com/pmanza/Cortical-D1D2.

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

## Acknowledgements
We thank Michele Vera-Yonga, Veronica Ramirez, Jamie Burns, Christopher Kure Liu, Dani Kroll, Dana Feldman, Karen Torres, Christopher Wong, Amna Zehra, Lori Talagala, Myke Vandine, and Minoo McFarland for their contributions. This work was accomplished with support from the National Institute on Alcohol Abuse and Alcoholism (Y1AA-3009).

## Author contributions
D.T., G.J.W., and N.D.V. conceived and designed the study. P.M., E.S.K., C.E.W., R.Z., N.G., K.M., E.B., E.D., and A.J. acquired the data. P.M., E.S.K., Ş.B.D., R.Z., E.D., A.J., D.T., and N.D.V. performed data analysis and interpretation. P.M. wrote the first draft of the manuscript. All authors revised and gave final approval to the manuscript.

## Funding

## Competing interests
The authors declare no competing interests.
