## [Peer Review File · Communications Biology]

Reviewers' comments:

Reviewer #1 (Remarks to the Author):

In this manuscript, Dr. Manza and colleagues investigated how the balance between cortical D1R versus D2R signaling coordinates the activity and connectivity of functional networks in the human brain using PET and MRI imaging tools. This is very interesting topic, and the manuscript is well written. The following are my comments that may improve the manuscript.

1. What is the rationale to use sensory mouth and sensory hand to represent the sensorimotor cortex? Are these the only regions available in the sensorimotor cortex in the atlas the authors used?
2. fALFF measures spontaneous fluctuations in BOLD-fMRI signal intensity. The actual significance of fALFF is still unknown. So, it may be fair to say that fALFF reflects low-frequency oscillation, and it may not be appropriate to claim fALFF represents "activity" of certain brain regions.
3. The authors tested whether the relative D1R/D2R ratio was significantly associated with age and spatial working memory performance, but it is not clear if the p values were adjusted for multiple tests? If the authors believe these are exploratory tests and p value adjustment is not needed, they should emphasize the exploratory characteristics of these tests.
4. Also, out of curiosity, how can gender and IQ influence the association between the D1R/D2R ratio and the age / working memory performance?
5. The authors mentioned that using two PET scanners is a limitation of the study. Can you explain why two scanners were used? Is this to speedup data collection or is this manuscript based on two experiments using identical design?

Reviewer #2 (Remarks to the Author):

The manuscript by Manza and colleagues report PET (targeting D1 and D2 receptors) and fMRI data at 3T recorded in separate sessions in healthy volunteers over a large age range in combination with a pharmacological challenge (methylphenidate). These data are novel and of high importance for the field. Data will be after publication publicly available for scientific use. The supplementary data comparison and discussion of raclopride and fallypride is convincing. Given the possibility of simultaneous MR-PET the authors might add a paragraph how a pharmacological challenge could benefit from this in the future. In summary, this manuscript should be considered for publication with highest recommendation.

Response to Reviewers

We thank the reviewers for their helpful comments. Our point-by-point responses are listed below.

Reviewer #1:

1. What is the rationale to use sensory mouth and sensory hand to represent the sensorimotor cortex? Are these the only regions available in the sensorimotor cortex in the atlas the authors used?

Response: Here we use all four of sensory mouth, sensory hand, visual cortex, and auditory cortex to represent the ‘sensorimotor’ systems. There are a number of nodes within each of the four networks that together provide good coverage of the sensory and motor cortices in the human brain. We chose to use this set of networks based on precedent, for comparability with many prior studies (e.g. Chan et al., 2017 J. Neurosci.) who defined association and sensorimotor cortices in this manner. The networks contributing to each system are represented in Figure 1B and described on **p. 8 lines 225-232** in the methods.

2. fALFF measures spontaneous fluctuations in BOLD-fMRI signal intensity. The actual significance of fALFF is still unknown. So, it may be fair to say that fALFF reflects low-frequency oscillation, and it may not be appropriate to claim fALFF represents “activity” of certain brain regions.

Response: Indeed, while many prior studies have argued that fALFF is a good proxy for “activity”, the exact significance of fALFF remains unknown. Studies using concurrent electrical recording of neuronal activity and fMRI in monkeys and rodents have found that the amplitude of BOLD, including during resting state, is strongly associated with gamma-band local field potential activity and to a lesser extent multiple-unit neuronal spiking activity (e.g., Goense & Logothetis, 2008 Current Biology; Magri et al., 2012 J. Neurosci; Scholvinck et al., 2010 PNAS). Nevertheless, the precise relationship between BOLD amplitude and neuronal activity remains under debate. We have added a statement to the limitations section **pp. 15-16 lines 461-468** describing this and cautioning the reader regarding the interpretation of ALFF as ‘activity’.

3. The authors tested whether the relative D1R/D2R ratio was significantly associated with age and spatial working memory performance, but it is not clear if the p values were adjusted for multiple tests? If the authors believe these are exploratory tests and p value adjustment is not needed, they should emphasize the exploratory characteristics of these tests.

Response: We did specifically hypothesize that significant correlations would be observed in association cortices based on an extensive literature implicating the association cortices in cognition and aging; therefore we did not perform multiple comparisons correction for these specific tests. We have now made explicit in the results section **pp. 12-13 lines 357 and 361** that these p-values are uncorrected. Of note, if Bonferroni-correcting for multiple comparisons the age relationship remains significant ($p = .01$) and the working memory relationship borders significance ($p = .05$).

4. Also, out of curiosity, how can gender and IQ influence the association between the D1R/D2R ratio and the age / working memory performance?

Response: We have added exploratory linear regression models (using the *lm* function in R) between Age and D1/D2R and between Working Memory and D1/D2R in association cortices where we include IQ and Sex as factors in the model. Neither factor was significant in either analysis (for the model with Age and D1D2R, the effect of sex: $t_{(35)} = -.185, p = .855$; and the effect of IQ: $t_{(35)} = 1.99, p = .055$; and for the model with spatial working memory and D1D2R, the effect of sex: $t_{(35)} = -.240, p = .812$; and the effect of IQ: $t_{(35)} = 1.500, p = .145$). In both models, the inclusion of these factors slightly reduced the strength of associations of Age and Spatial Working Memory with D1D2R, which is not surprising since performance on an IQ test should share some variance with age and cognitive test performance: (For the association of D1D2R and Age: $t_{(35)} = -2.154, p = .039$; and for the association of D1D2R and spatial working memory: $t_{(35)} = -1.774, p = .086$). We have added these results to the **Supplement**.

5. The authors mentioned that using two PET scanners is a limitation of the study. Can you explain why two scanners were used? Is this to speedup data collection or is this manuscript based on two experiments using identical design?

Response: The use of two scanners was necessary due to scheduling limitations at our site, so that we could complete data collection within a reasonable time frame. We have added this explanation to the methods **p. 6 lines 154-155**.

Reviewer #2:

1. Given the possibility of simultaneous MR-PET the authors might add a paragraph how a pharmacological challenge could benefit from this in the future.

Response: Thank you for this suggestion. We have now added a paragraph describing how simultaneous PET-pharmacological fMRI could give richer insights on the relationship between brain dopamine receptor signaling and functional activity on **p. 16 lines 469-480**.

References

- Chan, M. Y., Alhazmi, F. H., Park, D. C., Savalia, N. K., & Wig, G. S. (2017). Resting-State Network Topology Differentiates Task Signals across the Adult Life Span. *The Journal of Neuroscience*, 37(10), 2734–2745. <https://doi.org/10.1523/JNEUROSCI.2406-16.2017>
- Goense, J. B. M., & Logothetis, N. K. (2008). Neurophysiology of the BOLD fMRI Signal in Awake Monkeys. *Current Biology*, 18(9), 631–640. <https://doi.org/10.1016/j.cub.2008.03.054>
- Schölvinck, M. L., Maier, A., Ye, F. Q., Duyn, J. H., & Leopold, D. A. (2010). Neural basis of global resting-state fMRI activity. *Proceedings of the National Academy of Sciences of the United States of America*, 107(22), 10238–10243. <https://doi.org/10.1073/pnas.0913110107>

Magri, C., Schridde, U., Murayama, Y., Panzeri, S., & Logothetis, N. K. (2012). The amplitude and timing of the BOLD signal reflects the relationship between local field potential power at different frequencies. *Journal of Neuroscience*, 32(4), 1396–1407. <https://doi.org/10.1523/JNEUROSCI.3985-11.2012>

REVIEWERS' COMMENTS:

Reviewer #1 (Remarks to the Author):

The authors have addressed all of my comments.